# Development of a Droplet Digital PCR to Monitor SARS-CoV-2 Omicron Variant BA.2 in Wastewater Samples

**DOI:** 10.3390/microorganisms11030729

**Published:** 2023-03-12

**Authors:** Laura A. E. Van Poelvoorde, Corinne Picalausa, Andrea Gobbo, Bavo Verhaegen, Marie Lesenfants, Philippe Herman, Koenraad Van Hoorde, Nancy H. C. Roosens

**Affiliations:** 1Transversal Activities in Applied Genomics, Sciensano, 1050 Brussels, Belgium; 2Foodborne Pathogens, Sciensano, 1050 Brussels, Belgium; 3Epidemiology of Infectious Diseases, Sciensano, 1050 Brussels, Belgium; 4Biological Health Risks, Sciensano, 1050 Brussels, Belgium

**Keywords:** SARS-CoV-2, omicron, VOC, mutation, RT-ddPCR, variant detection, wastewater surveillance

## Abstract

Wastewater-based surveillance can be used as a complementary method to other SARS-CoV-2 surveillance systems. It allows the emergence and spread of infections and SARS-CoV-2 variants to be monitored in time and place. This study presents an RT-ddPCR method that targets the T19I amino acid mutation in the spike protein of the SARS-CoV-2 genomes, which is specific to the BA.2 variant (omicron). The T19I assay was evaluated both in silico and in vitro for its inclusivity, sensitivity, and specificity. Moreover, wastewater samples were used as a proof of concept to monitor and quantify the emergence of the BA.2 variant from January until May 2022 in the Brussels-Capital Region which covers a population of more than 1.2 million inhabitants. The in silico analysis showed that more than 99% of the BA.2 genomes could be characterized using the T19I assay. Subsequently, the sensitivity and specificity of the T19I assay were successfully experimentally evaluated. Thanks to our specific method design, the positive signal from the mutant probe and wild-type probe of the T19I assay was measured and the proportion of genomes with the T19I mutation, characteristic of the BA.2 mutant, compared to the entire SARS-CoV-2 population was calculated. The applicability of the proposed RT-ddPCR method was evaluated to monitor and quantify the emergence of the BA.2 variant over time. To validate this assay as a proof of concept, the measurement of the proportion of a specific circulating variant with genomes containing the T19I mutation in comparison to the total viral population was carried out in wastewater samples from wastewater treatment plants in the Brussels-Capital Region in the winter and spring of 2022. This emergence and proportional increase in BA.2 genomes correspond to what was observed in the surveillance using respiratory samples; however, the emergence was observed slightly earlier, which suggests that wastewater sampling could be an early warning system and could be an interesting alternative to extensive human testing.

## 1. Introduction

An unprecedented impact on global public health was observed due to the emergence of the severe acute respiratory syndrome coronavirus 2 (SARS-CoV-2) virus. Although there have been vaccines available since the end of 2020, new emerging variants may increase transmissibility, infectivity, and immune evasion, which could threaten global health again [1]. By monitoring the introduction and prevalence of new and existing variants of concern (VOC) and variants of interest (VOI) in a population, competent authorities can make better-informed public health decisions. Wastewater-based surveillance has already been successfully applied to monitor SARS-CoV-2 circulating in the community [2,3,4,5,6] and is an interesting complementary surveillance method to the SARS-CoV-2 surveillance using individual clinical samples, especially now that the amount of analyzed clinical samples is decreasing. Wastewater sampling provides objective information about virus circulation in a population, independently of the willingness and awareness of the person and diagnostic testing availability [7]. Methods for the detection of SARS-CoV-2 RNA in wastewater showed that the concentration in wastewater reflects and even precedes the trends seen in the SARS-CoV-2 surveillance using patient samples or hospitalizations [4,8]. Moreover, in case of a low SARS-CoV-2 prevalence, wastewater-based surveillance can be used as an early warning system [8,9,10].

Today, sequencing the SARS-CoV-2 genome is used more and more to monitor and identify the (new) lineages and mutations. However, polymerase chain reaction (PCR) assays remain often used for variant detection because they are more accessible and affordable as well as offering quantitative results [11,12,13]. Moreover, reverse transcriptase digital droplet PCR (RT-ddPCR) assays can analyze many samples in a few hours compared to sequencing which takes a much longer time [14]. However, a limitation of the RT-ddPCR assays is that they can only examine sequence fragments with a length of less than a hundred base pairs long [15]. If a mutation occurs in the targeted area, the assay will not work or not work efficiently, resulting in false negatives [5]. Moreover, an important limitation of PCR assays compared to sequencing is the inability to detect new mutations. Another drawback, which is even more problematic, is related to the fact that each SARS-CoV-2 variant is defined by a group of different mutations distributed across the genome, whereas the RT-ddPCR assay targets merely a hundred base pair fragments [16]. Therefore, due to the evolutionary relationship of different variants, they can often possess the same mutations, which reduces the specificity of the RT-ddPCR assays [16].

To tackle these challenges, it is key that SARS-CoV-2 genome databases are continuously monitored to evaluate and develop new PCR assays [17]. In a previous study, using all available whole-genome sequencing (WGS) data of SARS-CoV-2, we developed a multiplex RT-ddPCR method targeting all known SARS-CoV-2 variants [5]. Digital PCR technology was selected because this technology is very efficient at low virus concentrations, it is less sensitive to inhibition, and it allows absolute quantification [18,19]. Therefore, this technology is very convenient for the monitoring of wastewater [5]. A similar genomic approach can be used to develop ddPCR assays targeting specific mutations of virus variants in wastewater. The detection of single nucleotide mutations is often challenging; therefore, larger changes in the viral genome, such as the S-gene target failure (SGTF), are often used to identify both the B.1.1.7 and BA.1 variants [20,21]. Although SGTF identification could be useful for surveillance purposes, it is not specific to one variant [20,21]. Using ddPCR methods targeting all SARS-CoV-2 variants, on the one hand, and methods targeting a specific virus variant, on the other hand, might be a very advantageous approach. It allows the measurement of the proportion of a specific circulating variant in comparison to the total viral population.

In the present study, we developed an RT-ddPCR method that targets the T19I amino acid mutation in the spike protein, which is characteristic of the strain BA.2 that emerged in Belgium at the beginning of January 2022 and was gradually replaced by the emergence of BA.4 and BA.5, which also possess the T19I mutation, at the end of May 2022. First, publicly available whole-genome sequencing data were used to evaluate the primers and probes regarding a large Global Initiative on Sharing All Influenza Data (GISAID) dataset. Subsequently, the primers and probes were tested in vitro for their specificity and sensitivity and the sensitivity of the T19I assay. By using this specific method, that uses a mutant and wild-type to quantify both populations, the proportion of genomes with the T19I mutation, characteristic of the BA.2 mutant, can be calculated. Finally, in this context, wastewater samples were used as a proof of concept to monitor the emergence and the prediction and scaling of the spread of the BA.2 variant over time.

## 2. Materials and Methods

### 2.1. Selection and Evaluation of a BA.2 Target Using Whole-Genome Sequencing Data

In collaboration with IDT (Integrated DNA Technologies, Coralville, IA, USA), two locked nucleic acid (LNA) probes were designed for the differentiation between the wild-type (WT) and the mutation at position 19 in the S protein (T19I = C21618T) present in BA.2. Different fluorophores were conjugated to the 5′ end of each TaqMan probe (HEX and FAM, respectively) to facilitate the differentiation of the respective fluorescence signals. LNA probes have the advantage of having short sequences that improve mismatch discrimination and increased affinity for their complementary strand [22]. Additionally, because of the competition between the WT and mutant probe, the specificity of the mutant probe to detect the BA.2 will increase while the false positive results will decrease.

The in silico evaluation of the specificity of the three assays (Table 1) was carried out using an in-house-developed R script using R-software (RStudio 4.2.0; R3.6.1) that used recent whole-genome SARS-CoV-2 sequences. A total of 14,422,827 SARS-CoV-2 genomes, obtained from samples collected between 24 December 2019 and 15 December 2022, were obtained from the GISAID database [23] on 29 December 2022. Genomes containing undetermined nucleotides “N” and degenerate nucleotides were excluded from the dataset to retain only high-quality genomes. This results in 4,593,520 genomes, of which 342,046 were BA.2 genomes (Appendix A). Moreover, a subset of this dataset was taken based on the sampling date that ranges from week 4 (24 January 2022) to week 20 (22 May 2022) of 2022, corresponding to the samples chosen to show the emergence of BA.2 in Belgium (840,419 sequences, of which 301,613 were BA.2 genomes). For each SARS-CoV-2 genome that was analyzed, a negative detection signal was defined by the presence of at least one mismatch, which is defined as a theoretical false-negative result.

### 2.2. Development of RT-ddPCR Method for the Detection of the SARS-CoV-2 Variant BA.2

The RT-ddPCR assay was evaluated using purified RNA from the SARS-CoV-2 variant BA.2 (Vircell, Granada, Spain–MBC145-R). The One-Step RT-ddPCR Advanced Kit for Probes (Bio-Rad, Hercules, CA, USA) was used to perform the RT-ddPCR. All kit components were thawed on ice for 30 min and thoroughly mixed by vortexing the tubes for 30 s at maximum speed. A larger master mix was produced with the reagents and subsequently aliquoted into individual reactions. Each reaction, set up on ice, had a total volume of 22 µL, including 0.55 µL for the T19I assay and 0.99 µL for the RdRp and ORF1a assays of each primer, with an initial concentration of 20 µM and 0.44 µL for the T19I assay and 0.55 µL for the RdRp and ORF1a assays of each probe, and 0.99 µL of each primer with an initial concentration of 10 µM, 1.1 µL of 300 mM DTT, 5.5 µL One-Step Supermix, 2.2 µL Reverse Transcriptase, 8 µL of sample, and a certain volume of dH2O to achieve a total volume of 22 µL (Table 2). According to the manufacturer’s instructions, 20 µL of the reaction mix and 70 µL of Droplet Generation Oil for Probes were loaded into a QX200TM droplet generator (Bio-Rad), and to increase the number of droplets, the cartridge was kept for two minutes at room temperature. After the droplet generation, 40 µL of droplets was recovered per reaction. The amplification was performed in a T100TM Thermal Cycler (Bio-Rad) with the following conditions: one cycle at 25 °C for 3 min, one cycle at 50 °C for 60 min (RT), one cycle at 95 °C for 10 min (Taq polymerase activation), 40 cycles at 95 °C for 30 s (denaturation), 57.5 °C for 60 s (annealing), one cycle at 98 °C for 10 min (enzyme inactivation), and finally one cycle at 4 °C for 30 min (stabilization). Next, the plate was transferred to the QX200 TM reader (Bio-Rad) and the results were acquired using the HEX and FAM channels, according to the manufacturer’s instructions. The QuantaSoft software v1.7.4.0917 (Bio-Rad) was used for the interpretation of the results and the threshold was set manually.

### 2.3. Validation of In Vitro Sensitivity of RT-ddPCR Assay for BA.2

The sensitivity was evaluated by using serial dilutions of purified RNA from the SARS-CoV-2 BA.2 virus (Vircell, Granada, Spain—MBC145-R). A negative control (dH2O) and seven serial dilutions were prepared, starting at an average concentration of 71.61 ± 5.28 copies/µL and serially diluted at 2×, 5×, 10×, 20×, 50×, and 100×, and each dilution was tested in 12 replicates. The limit of blank (LOB) was defined as the upper 95% confidence limit of the mean false-positive measurements using the number of droplets in all negative samples using the following formula [26]:𝜇𝑐𝑜𝑟𝑟(droplets) = 𝜇(droplets) + 1.645 σ(droplets) (𝜇 = mean; σ = standard deviation)

In this study, we opted for the calculation of the LOB95% to exclude false positives. The LOB95% of the T19 and I19 probes was estimated to be 3 and 2 droplets, respectively, which means that there is a 95% chance that if more droplets are detected than the LOB95%, it is not a false-positive result. Other studies use, for example, an arbitrary limit of 9 droplets [27], which would also take into account the false positives but will likely result in more false-negative results. These negative samples included the negative controls of the sensitivity test and all samples of the specificity test except BA.2 for the I19 target. For the T19 target, the negative controls of the sensitivity test and all samples except the SARS-CoV-2 variants besides BA.2 were used. Using the web application Quodata [28], the limit of detection (LOD95%) was calculated with the number of copies of the target that are required to ensure a probability of detection (POD) of 95%.

### 2.4. Validation of In Vitro Specificity of RT-ddPCR Assay for BA.2

DNA and RNA controls were used to experimentally validate the specificity of the method. A list of these controls can be found in Table 5 or Appendix A. Each material was tested in duplicate and included 500 copies/µL for the viruses, while the bacterial, fungal, plant, and human DNA contained 0.5 ng/µL.

### 2.5. A Proof of Concept for the Monitoring of Virus Variants in Wastewater

After the validation of the T19I assay on RNA controls, the assay was also evaluated using non-artificial samples. Every two weeks starting in week 4 of 2022, two samples from the two wastewater treatment plants covering the entire Brussels-Capital Region, Belgium, were collected, which resulted in 18 wastewater samples. These samples were selected based on the genomic surveillance in respiratory samples to include the weeks of the emergence of the BA.2 variant in Belgium. Both the T19I assay and the assay to detect all SARS-CoV-2 variants, targeting the ORF1a and RdRp proteins [5], were used to evaluate their applicability to these wastewater samples. The collection and extraction of these samples were performed as described previously in Van Poelvoorde et al. [5].

## 3. Results

### 3.1. In Silico Inclusivity Evaluation for the T19I, ORF1a, and RdRp Assays

The inclusivity of the T19I, ORF1a, and RdRp assays was evaluated with a dataset of 4,593,520 GISAID genomes, of which 342,046 were BA.2 genomes. The ORF1a and RdRp assays are part of the general assay that was previously validated to detect the total amount of SARS-CoV-2 in a sample [5]. Excellent inclusivity was obtained for the ORF1a and RdRp assays (Table 3). Moreover, within the BA.2 genomes, high inclusivity was obtained for the ORF1a and RdRp assays (Table 3). The little variation observed can mainly be attributed to random and rare mutation events that did not spread in the viral population. Moreover, when these assays were evaluated on the genomes that were sampled between week 4 (24 January 2022) and week 20 (22 May 2022) 2022, which corresponds to the samples chosen to show the emergence of BA.2 in Belgium, the inclusivity of the primers and probes remain high (840,419 sequences of which 301,613 were BA.2 genomes). Consequently, these two assays proved reliable assays to detect all SARS-CoV-2 viruses.

While the general assays should target all SARS-CoV-2 genomes, the T19I assay targets the variant BA.2 specifically. The primers of the T19I assay show great inclusivity for all SARS-CoV-2 sequences (Table 3). The T19I assay showed that the mutant probe exhibits excellent inclusivity for BA.2 genomes (Table 3), while there are only a few BA.2 genomes that match with the wild-type probe. The inclusivity of the wild-type probe is relatively low, 36.96% of the sequences could not be detected with either the mutant or wild-type probe, because of certain mutations related to specific variants. From the VOCs, sequences belonging to the variants P.1 and B.1.617.2 have two mutations (C21614T and C21621A) and one mutation (C21618G) in the target region, respectively. However, both of these lineages were not circulating at that time. When the T19I assay was evaluated on the genomes in the period ranging from week 4 to week 20 in 2022, then the inclusivity of both the forward and reverse primer remains high and only 0.49% of the genomes could not be identified by either the mutant or the wild-type probe (Table 3). The exclusivity of the mutant probe compared to other SARS-CoV-2 is high. Indeed, this mutation allows BA.2 and all its subvariants to be differentiated from all other SARS-CoV-2 variants including BA1.

### 3.2. In Vitro Sensitivity Assessment of the Mutant I19 Assay

Serial dilutions ranging from 0.5 to 50 copies/µL of the SARS-CoV-2 BA.2 control were used to estimate the sensitivity of the I19 mutant probe. The LOB_95%(MUT)_ for the mutant probe was estimated at two droplets, while the LOB_95%(WT)_ for the wild-type probe was estimated at three droplets. If the number of droplets was below these LOB thresholds, they were considered negative. Amplification for all 12 replicates was observed until 6.19 target copies/µL was obtained (Table 4). The LOD_95%_ for the I19 mutant probe was estimated at 3.576 target copies/µL.

### 3.3. In Vitro Specificity Assessment of the T19I Assay

The T19I assay was experimentally evaluated for each positive and negative material (Table 5). RNA from the BA.2 variant was used as a positive control, while six other SARS-CoV-2 variants, four closely related coronaviruses, ten other viruses, and DNA from a plant, two bacteria, two fungi, and a human were used as negative controls. The positive control presented an amplification, while all negative controls tested below the LOD.

**Table 5 microorganisms-11-00729-t005:** Specificity assessment of the T19I assay. The absence and presence of amplification are symbolized by a “−“ or “+”, respectively. The RT-ddPCR method was performed in duplicate on each sample. As positive control SARS-CoV-2 RNA from the BA.2 variant was included.

Kingdom	Genus	Species	Strain Number	I19 Mutant
Animalia	*Homo*	*sapiens*	/	−
Plantae	*Zea*	*mays*	/	−
Bacteria	*Bacillus*	*subtilis*	SI0005	−
	*Escherichia*	*coli*	MB1068	−
Fungi	*Aspergillus*	*acidus*	26285	−
	*Candida*	*cylindracea*	041387	−
	**Family**		**Species**	**I19 Mutant**
Viruses	*Picornaviridae*		Rhinovirus B	−
	*Reoviridae*		Rotavirus	−
	*Orthomyxoviridae*		Influenza A (H1N1)	−
	*Orthomyxoviridae*		Influenza A (H3)	−
	*Orthomyxoviridae*		Influenza B	−
	*Adenoviridae*		Adenovirus	−
	*Picornaviridae*		Enterovirus D68	−
	*Caliciviridae*		Norovirus	−
	*Pneumoviridae*		RSV A	−
	*Coronaviridae*		SARS-CoV	−
	*Coronaviridae*		MERS-CoV	−
	*Coronaviridae*		Corona OC43	−
	*Coronaviridae*		Coronavirus control	−
	*Coronaviridae*		SARS-CoV-2 WT	−
	*Coronaviridae*		SARS-CoV-2 B.1.1.7	−
	*Coronaviridae*		SARS-CoV-2 B.1.351	−
	*Coronaviridae*		SARS-CoV-2 P.1	−
	*Coronaviridae*		SARS-CoV-2 B.1.617.2	−
	*Coronaviridae*		SARS-CoV-2 BA.1	−
	*Coronaviridae*		SARS-CoV-2 BA.2	+

### 3.4. A Proof of Concept for the Monitoring of Virus Variants in Wastewater

The presence and quantity of the BA.2 variant and SARS-CoV-2, in general, were assessed in eighteen wastewater samples of the two wastewater treatment plants of Brussels-Capital Region, capital of Belgium. These 18 wastewater samples were collected every 2 weeks starting in week 4 of 2022 to evaluate the applicability of wastewater surveillance to monitor the emergence of VOCs. Thanks to the use of the mutant (I19) and the wild-type (T19) probes, the proportion of the BA.2 variant in wastewater was determined. These results were compared to the publicly available data from the Belgian genomic surveillance performed by the Belgian Sequencing Consortium, which includes clinical samples collected across Belgium [29] (Figure 1).

After the emergence of variants containing the T19I mutation in week 7 in the epidemiological clinical data, the increase in the proportion of this mutation in wastewater follows a trend that is comparable with the increase in T19I-containing variants in patients. The emergence of the I19 mutation can be seen a little bit earlier (week 6) compared to the epidemiological clinical data. At approximately week 16, BA.2 reached a plateau, which means that almost all SARS-CoV-2 cases in Brussels-Capital Region were due to the BA.2 variant. It should be noted that there is a drop in week 14 in station 36A. Potentially, this could be due to heavy rainfall as there was a peak in the flow rate during week 14 at station 36A.

## 4. Discussion

This study describes the development of a new RT-ddPCR method to specifically detect and quantify the amino acid mutation T19I associated with the BA.2 variant using two LNA probes that target the mutant and wild-type. The use of this kind of ddPCR method allows us to monitor but also quantify the proportion of SARS-CoV-2-variant-carrying SARS-CoV-2 mutations associated with VOCs in comparison with the entire SARS-CoV-2 population in wastewater. Regarding the development of the T19I method, to properly assess the specificity of the method, the assay should ideally be tested against a large number of various SARS-CoV-2 strains. However, it is difficult to obtain and experimentally test a representative collection of all circulating strains [5]. Therefore, a total of 4,593,520 high-quality genomes, including 342,046 BA.2 genomes, were used to evaluate the in silico inclusivity of the newly developed T19I assay targeting the BA.2 variant. Additionally, the ORF1a and RdRp assays that were previously developed and that targeted all SARS-CoV-2 variants were re-evaluated in the context of the new variant. Excellent results were obtained for the T19I, ORF1a, and RdRp assays. Inclusivity of more than 97% was obtained for the detection of all SARS-CoV-2 viruses using the ORF1a and RdRp assays, while 99.70% of the BA.2 genomes match with the I19 mutant probe and 0.12% of the BA.2 genomes match with the T19 wild-type probe. Consequently, the general assay, including the ORF1a and RdRp assays, seems to still be an appropriate assay to detect all SARS-CoV-2 viruses, despite the circulation of many variants. Moreover, the inclusivity of the primers and probes from the T19I assay confirms that it is a suitable and specific assay to detect BA.2 genomes. The exclusivity of the T19I assay is mostly limited to BA.2 and its subvariants, which makes it an appropriate target to detect the emergence of BA.2 during the decline of BA.1. However, the T19I mutation could also be found in BA.4 and BA.5 variants that are closely related to the BA.2 variant (sharing 23 characteristic mutations) and eventually replaced the BA.2 variant during the spring of 2022. Consequently, this mutation remained present in the circulating variants and, therefore, this assay could not be used to observe the potential decrease in the presence of this variant and the emergence of BA.4 and BA.5.

Following the in silico evaluation of the T19I assay, the performance of the developed method was evaluated using a minimal experiment set-up as previously described in Van Poelvoorde et al. [5]. First, the sensitivity of the method was evaluated. The LOD_95%_ of the I19 probe was estimated to be 3.576 copies/µL. There are a few other studies that detect the BA.2 variant. Mills et al. [30] provides multiple assays and the combination of mutations described leads to the conclusion of which variant is present in the respiratory sample. The observed LOD ranging from 6.5 to 14 copies/reaction (10 µL sample/reaction) was slightly more sensitive, while they needed to observe at least three positive droplets before the result would be accepted. Nevertheless, it is not possible to use a combination of mutations to characterize variants in wastewater because multiple variants can be present. Subramoney et al. [16] used the spike gene target failure to distinguish the BA.2 variant. However, the spike gene target failure is not a specific mutation for BA.2, so especially in wastewater samples that contain a mix of SARS-CoV-2 variants it is likely that there are also other variants present with this deletion. Moreover, as no LOD was defined, the performance results could not be compared [16]. Second, the specificity of the T19I assay was evaluated using a set of DNA and RNA references. No false-positive results were observed above the LOB_95%_ for human, plant, bacterial, or viral DNA and RNA, including closely related viruses, such as Corona OC43, MERS-CoV, and SARS-CoV, and other SARS-CoV-2 variants than BA.2. The T19I assay has the advantage of being able to detect BA.2 specifically instead of using a combination of mutations, which is not possible when characterizing wastewater samples. Based on the inclusivity results, we can conclude that almost all of the BA.2 genomes can be detected using the T19I assay, while the general assays are still able to detect more than 97% of all SARS-CoV-2 viruses. Such method performance regarding the specificity and sensitivity indicates that the developed method should be adequate for robust absolute quantification of SARS-CoV-2 in wastewater samples that may have low concentrations of SARS-CoV-2.

Third, RNA extracted from wastewater samples was used as a proof of concept to monitor a specific variant over time in a population. The samples were selected based on the genomic surveillance in respiratory samples to include the week of the emergence of the BA.2 variant. Although the number of selected wastewater samples was limited, the same trend in the proportion of BA.2 was observed in wastewater compared to the surveillance in individuals, although the emergence of the BA.2 variant can be detected approximately a week earlier compared to the epidemiological clinical data. This seems to correspond to other papers that reported wastewater surveillance as an early warning system. It should be noted that the variants BA.4 and BA.5 appeared in the spring of 2022. This is not detectable using the developed RT-ddPCR method, which is targeting a mutation common to these variants. Such an analysis could only be performed retrospectively because the sequence of all the variants should be available to identify unique mutations. In particular, since the emergence of the omicron variant, this has become challenging because the difference between each variant is limited to only a few mutations. For example, there is only one unique BA.2 mutation compared to BA.1, BA.4, and BA.5.

Wastewater surveillance has been a very useful surveillance tool to monitor the spread of SARS-CoV-2 on a population level and has been instrumental in providing early warnings in care facilities and remote communities [31]. The assay used in this study provides sufficiently high sensitivity and specificity for the detection and quantification of the T19I mutation in wastewater. Moreover, this method allows the calculation of the proportion of mutants and wild-types that are circulating in the population. This assay consists of two individual RT-ddPCR reactions that can easily be integrated into the current wastewater surveillance protocols for cost-effective and rapid detection and quantification of mutations associated with SARS-CoV-2 BA.2. Moreover, the T19I assay was used to test wastewater samples from a densely populated area such as Brussels-Capital Region to determine the relative BA.2 proportion. An increase in the proportion of variants containing the T19I mutation was observed during the spring of 2022, which corresponds to the observations in the genomic surveillance of respiratory samples. The results obtained by this quantitative method indicate that this strategy could be useful to predict and scale the spread of new VOCs. Consequently, wastewater surveillance could be an interesting cost-effective warning system for communities, and this study demonstrates the value of RT-ddPCR in detecting T19I. Moreover, this strategy can also be used to detect other signature mutations characteristic of relevant VOCs. In a transition period between two variants, one could also consider removing the fluorophore of the WT probe to combine this with another assay targeting the second variant. This assay is simpler, cheaper, and faster than whole-genome sequencing, which is the gold standard to detect variants. Furthermore, the format of this method is readily adaptable to additional emerging VOCs by changing the primers and probes as the need arises.

Although RT-ddPCR remains a very convenient tool, this study illustrates also that the early use of whole-genome sequencing in wastewater samples remains indispensable in the early detection of new emerging variants and in case the need arises to confirm multiple mutations. Moreover, whole-genome sequencing is essential initially to allow the development of new methods. However, once the RT-ddPCR method is developed, wastewater surveillance can be carried out cheaper and faster.

## Figures and Tables

**Figure 1 microorganisms-11-00729-f001:**
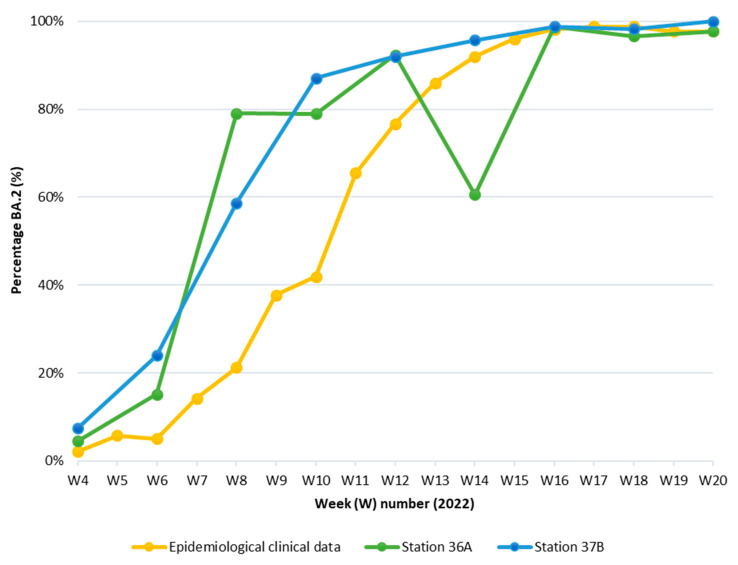
The concentration of mutant I19 gene divided by the total amount of virus that was found with the wild-type T19 and mutant I19 target together in two wastewater treatment plants (green and blue lines) in Brussels-Capital Region from week 4 to week 20 2022 compared to genomic surveillance data obtained by the Belgian Sequencing Consortium (yellow line). Station 36A covers a population of approximately 1,045,863 inhabitants in the north of Brussels and Station 37B covers a population of approximately 311,866 inhabitants.

**Table 1 microorganisms-11-00729-t001:** Primers and probes that target the T19I variant and the general assays targeting the RdRp [24] and ORF1a [25] genes. A second, internal ZEN-quencher was added to ORF1a and RdRp probes to obtain greater overall dye quenching in addition to the Iowa Black FQ (IABkFQ) quencher. The indicated positions refer to the reference sequence NC_045512. The “+” within the sequences indicates the LNA nucleotides within the LNA probe.

Target	Primer/Probe	Sequence	Nucleotide Position	Final Concentration	Amplicon Length (bp)
T19I	T19I_FW	TTATTGCCACTAGTCTCTAGTCA	21,581–21,603	0.5 µM	96
	T19I_RV	GGTAATAAACACCACGTGTGAA	21,656–21,677	0.5 µM	
	T19I_MUT	FAM/CT+T+A+T+AA+C+CA+GAA/IABkFQ	21,614–21,626	0.2 µM	
	T19I_WT	HEX/CTT+A+C+AA+C+C+AGAA/IABkFQ	21,614–21,626	0.2 µM	
ORF1a	ORF1a-F	AGAAGATTGGTTAGATGATGATAGT	3193–3217	0.9 µM	117
	ORF1a-R	TTCCATCTCTAATTGAGGTTGAACC	3286–3310	0.9 µM	
	ORF1a-P	FAM/TCCTCACTG-ZEN-CCGTCTTGTTGACCA/IABkFQ	3229–3252	0.25 µM	
RdRp	RdRp_IP4-F	GGTAACTGGTATGATTTCG	14,080–14,098	0.9 µM	106
	RdRp_IP4-R	CTGGTCAAGGTTAATATAGG	14,167–14,186	0.9 µM	
	RdRp_IP4-P	HEX/TCATACAAA-ZEN-CCACGCCAGG/IABkFQ	14,105–14,123	0.25 µM	

**Table 2 microorganisms-11-00729-t002:** Composition of the mixture for the RT-ddPCR reaction.

Components	Initial Concentration	Volume (µL)
T19I_FW	20 µM	0.55
T19I_RV	20 µM	0.55
T19I_MUT	10 µM	0.44
T19I_WT	10 µM	0.44
DTT	300 nM	1.1
dH_2_O		3.22
Reverse transcriptase		2.2
Supermix	2X	5.5
**Master mix**		14
RNA template		8
**Total**		22

**Table 3 microorganisms-11-00729-t003:** Inclusivity in silico evaluation of T19I, ORF1a, and RdRp assays. For the two datasets, all sequences and sequences after 1 December 2021 were evaluated in silico for the T19I, ORF1a, and RdRp assays. The inclusivity shows the percentage of genomes that perfectly matched the primer or probe. Moreover, this was calculated for only the genomes belonging to the BA.2 lineage. The false negatives (FN) are the number of genomes that did not perfectly match with the primer or probe.

	All Sequences (4,593,520)			Sequences between W4 and W20 2022 (840,419)		
Primer/Probe	Inclusivity	FN	Inclusivity BA.2	Inclusivity	FN	Inclusivity BA.2
T19I_FW	96.30%	170,144	99.81%	91.14%	74,453	99.83%
T19I_RV	99.52%	21,991	99.58%	99.81%	1614	99.76%
T19I_WT	36.96%	2,895,673	0.17%	37.97%	521,293	0.12%
T19I_MUT	29.49%	3,238,711	99.61%	61.54%	323,265	99.70%
ORF1a-F	99.80%	9062	99.87%	99.85%	1242	99.87%
ORF1a-R	99.63%	16,994	99.74%	99.83%	1437	99.74%
ORF1a-P	98.70%	59,301	99.48%	97.86%	17,958	99.48%
RdRp_IP4-F	99.88%	5369	99.82%	99.90%	808	99.83%
RdRp_IP4-R	99.27%	33,669	99.51%	99.46%	4577	99.55%
RdRp_IP4-P	97.63%	108,643	99.42%	98.75%	10,478	99.44%

**Table 4 microorganisms-11-00729-t004:** Sensitivity assessment of the I19 mutant probe using the SARS-CoV-2 BA.2 variant control. The absence or presence of amplification is indicated by − or +, respectively. For each average concentration (±the standard deviation), 12 replicates were tested, and the number of positive replicates is indicated between brackets at the middle line of each box.

Average Concentration (STDEV)	Sensitivity Assessment (I19 = Mutant)
71.61 ± 5.28 copies/µL	+ (12/12)
36.55 ± 3.14 copies/µL	+ (12/12)
13.25 ± 2.11 copies/µL	+ (12/12)
6.19 ± 1.64 copies/µL	+ (12/12)
2.29 ± 0.67 copies/µL	+ (10/12)
1.06 ± 0.46 copies/µL	+ (7/12)
0.83 ± 0.11 copies/µL	+ (6/12)
0 copies/µL	− (0/12)

## Data Availability

The data presented in this study are available within this article or in the Appendix A.

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
