# Peer review of "Development of a Droplet Digital PCR to Monitor SARS-CoV-2 Omicron Variant BA.2 in Wastewater Samples"

_microorganisms, 2023, doi:10.3390/microorganisms11030729_

Round 1

Reviewer 1 Report

Title: Identification of SARS-CoV-2 Omicron variant BA.2 using droplet digital PCR in wastewater samples

Type: research article

General comment: In this work a highly sensitive ddPCR method was developed both in silico and experimentally for detecting and quantifying SARS-CoV-2 Omicron variant BA.2 in wastewater samples. Upon a huge computational analysis using a dataset of all known SARS-CoV-2 genomes, a small number of wastewater samples were used as a proof of concept to monitor the emergence of the BA.2 variant over time. In my opinion, the present manuscript could improve our knowledge in the field of highly sensitive methods for detecting SARS-CoV-2 infection, with obvious important applications. Enclosed please see some specific comments alongside minor suggestions to further improve this interesting work, which I recommend for publication in microorganisms MDPI.

Comment 1. I suggest reducing the introduction by 20% in order to improve its readability. For instance, the study aim can be shortened.

Comment 2. A s ageneral comment, I suggest including more supporting references in the introductive and methods sections

Comment 3. I suggest improving the conclusions by emphasizing the reliability of the described ddPCR method

Commen 4. An important study limitation is the limited number of wastewater samples used for assay validation. This should be clearly underlined in the discussion

Minor comments

1)      Line 48 reference? For instance https://doi.org/10.3390/v14112351

2)      This recently published comprehensive review on the most effective molecular methods for detecting SARS-CoV-2, which also describes the droplet digital PCR method, should be included DOI: 10.3390/microorganisms10061193, e.g., introduction, lines 53-55

3)      More references should be included in lines 58-73 as a support of the statements.

4)      In lines 116-117 the is a typo error. Same comment in lines 211-213, 233-234, 251, 259, 276

5)      Lines 155-156 is that a plasmid?

6)      Please include the source of the dataset mentioned in line 207

7)      Line 220 better “virus”

8)      In table 4, Coronaviridae should be in italics. In general, virus families should be in italics

Reviewer 2 Report

The article titled "Identification of SARS-CoV-2 Omicron variant BA.2 using droplet digital PCR in wastewater samples " elucidates the use of ddPCR to detect the Omicron BA.2 variant in wastewater, which has proven to be a valuable approach in comprehending the viral dissemination in the community during the pandemic. While ddPCR presents unique advantages for viral detection in wastewater, the need for further studies to explore its application is warranted. However, still major concerns need to be better addressed:

1.       This study developed and assessed the T19I assay for discerning between BA.2 and wild type. However, the presence of T19I mutations in multiple subsequent Omicron variants, including BA4, BA.5, and recently observed XBB, BF.7, has greatly restricted the applicability of the T19I assay. Consequently, this assay is currently only scientifically valuable for a limited time period, which reduced the significance of the study's findings. 

2.       The authors stated “it is not possible to use a combination of mutation to characterize variants in wastewater because multiple variants can be present”. Other studies had used combination of several mutations in ddPCR test in wastewater (see below) and showed that the combination could distinguish more variants. Could you elaborate on the differences and the reasons?

Caduff, Lea, et al. "Inferring transmission fitness advantage of SARS-CoV-2 variants of concern from wastewater samples using digital PCR, Switzerland, December 2020 through March 2021." Eurosurveillance 27.10 (2022): 2100806.

Boogaerts, Tim, et al. "Optimization and application of a multiplex digital PCR assay for the detection of SARS-CoV-2 variants of concern in Belgian influent wastewater." Viruses 14.3 (2022): 610.

Other comments:

1.     Line 46. What do you mean the “follow-up of individual clinical samples”?

2.     Currently, NGS has been widely used for the monitoring and identification of new lineages and mutations of SARS-CoV-2. It is better to include NGS method also in introduction part.

3.     In table 1, there are some “+” in T19I probe sequences. What is the meaning?

4.     Line 116, it showed “Error! Reference source not found”, and it happens many times in the article, please fix it.

5.     The authors performed an in silico evaluation of the assay in December 2022 (Line 119-120). Could you provide a rationale for performing the BA.2 assay during this time?

6.     Line 135-141. Please rewrite it and make it more clear. For example, “0.55 μL for the T19I assay and 0.99 μL for the RdRp and ORF1a assays of each primer with an initial concentration of 0.5 μM”. However, the concentration for RdRp and ORF1 primers are 0.9 μM in table 1.

7.     Line 223-225. “there are only few BA.2 genomes that match with the wild type probe (0.17%)”. Based on GISAID data, a small proportion of BA.2 and following variant do not have T19I mutation, this could be the reason.

8.     In table 2. All sequences are labelled 4 073 722, whereas I can’t find this number in materials and methods part.

9.     In table 3. The last row, “SARS-CoV-2” should also be “Coronaviridae”.

10.   Could you elaborate the reasons to add humans, plant, fungi in the specificity assessment?  

11.   In Figure 1. Station 36A had a drop of BA.2 gene in week 14. Do you have an explanation of possible reason?

12.   In discussion, the authors stated that ddPCR is a cheaper and faster method in wastewater surveillance. In consideration of time and reagent price, it is really cheaper and faster than qPCR?

Reviewer 3 Report

The manuscript describes the evaluation of an assay for detecting the BA.2 variant (Omicron sublineage) of SARS-CoV-2 virus by identifying the T19I mutation using droplet digital PCR.  The analysis was performed on samples collected from wastewater to determine its suitability for use in wastewater-based surveillance. The authors determined, both in silico and in vitro, its level of inclusivity, sensitivity and specificity.

Although the applied approach is interesting and epidemiologically important, the manuscript, especially the results, are presented in a chaotic manner, making it difficult for the reader to follow the presented research. In conclusion, in my opinion, the manuscript needs major revisions to be published.

1.     the first time the names used must be full, and only then can abbreviations be used (e.g. line 39: SARS-CoV-2; line 55: RT-ddPCR assays),

2.     lines 53-55: please add more references (e.g.: https://doi.org/10.1371/journal.pone.0269071; https://doi.org/10.3390/ijms23169416),

3.     information in lines 58-66 should rather be included in the discussion,

4.     information in lines 84-88 should rather be included in the discussion,

5.     lines: 116, 211, 213, 234, 251, 259, 276: “Error! Reference source not found” should be removed,

6.     information contained in “Development of RT-ddPCR method for the detection the SARS-CoV-2 variant BA.2” should be present in a table,

7.     lines 166-168: information on the results of other published studies should be included in the discussion, not in Materials and Methods. It should also be explained in the discussion on what basis the Authors assumed that 2-3 drops are sufficient for the reliability of the study, although other scientists recommend 9. Does this mean that for SARS-CoV-2 you would rather have more false positives than false negatives? Is the method supposed to be sensitive or specific?,

8.     For the information contained in “Validation of in vitro specificity of RT-ddPCR assay for BA.2”, please insert in table,

9.     in general to the results: present only facts, all observed relationships should be included in the discussion,

10.  lines 207-238: lease shorten; do not include information that is included in the tables,

11.  please add a limitation to the study.

Reviewer 4 Report

This study is a paper that conducted a study to confirm BA.2 by applying ddPCR in terms of methods used in monitoring wastewater-based epidemiology, and is an important study to confirm sensitivity and specificity in terms of methods.

However, presenting a very important methodological opinion in the study of wastewater-based epidemiology has provided a very important issue. However, it is considered a very narrow study to identify only the BA.2 detailed variation, which is the Omicron variant.

It would be very valuable if the title and contents were changed and expanded to the development and application of the ddPCR method for wastewater-based dynamics.

Please attach the results and discussions that have been expanded a little more just by analyzing the analysis results of mutant stocks using specific SNPs.

Round 2

Reviewer 2 Report

The authors have responded the concerns from previous version and those changes signifcantly improve the quality of the manuscript. 

A small issue: In newly added table 2, the whole volume of the ddPCR system is 22 ul, if you use 2X supermix, should it be 11 ul instead of 5.5 ul?

Author Response

Thank you for your reviewing and for your valuable contributions and indeed, we thank you for pointing out this error. The total should be 22 µL. The concentration shows the initial concentration and this has now been adapted accordingly. 

Reviewer 3 Report

The manuscript can be published in its present form. Congrats!

Author Response

Thank you for your reviewing and for your valuable contributions.

Reviewer 4 Report

This study presented a new method for monitoring the mutant virus of Covid-19 based on wastewater. However, it would be a very important study to suggest a clearer and new method by suggesting that it can be applied by predicting and scaling the spread of the newly identified Varints of Concern virus rather than by identifying only BA.2 in the current research topic.

Author Response

We thank you for your reviewing and for your valuable contributions. As suggested we have adapted the introduction and discussion part of the manuscript to reflect better the advantage of this method to predict and scale the spread of VOCs.